# Alcohol Control Policy in Europe: Overview and Exemplary Countries

**DOI:** 10.3390/ijerph17218162

**Published:** 2020-11-04

**Authors:** Nino Berdzuli, Carina Ferreira-Borges, Antoni Gual, Jürgen Rehm

**Affiliations:** 1WHO Regional Office for Europe, UN City, Marmorvej 51, 2100 Copenhagen, Denmark; Berdzulin@who.int; 2WHO European Office for Prevention and Control of Noncommunicable Diseases, Leontyevsky Pereulok 9, 125009 Moscow, Russia; ferreiraborgesc@who.int; 3Clinical Addictions Research Group (GRAC-GRE) Psychiatry Department, Neurosciences Institute, Hospital Clínic, University of Barcelona, C/Mallorca 183, 08036 Barcelona, Spain; TGUAL@clinic.cat; 4Institut d’Investigacions Biomèdiques August Pi i Sunyer (IDIBAPS), C/Mallorca 183, 08036 Barcelona, Spain; 5Institute for Mental Health Policy Research, Centre for Addiction and Mental Health (CAMH), 33 Ursula Franklin Street, Toronto, ON M5S 2S1, Canada; 6Campbell Family Mental Health Research Institute, CAMH, 250 College Street, Toronto, ON M5T 1R8, Canada; 7Department of Psychiatry, University of Toronto, 250 College Street, 8th Floor, Toronto, ON M5T 1R8, Canada; 8Dalla Lana School of Public Health, University of Toronto, 155 College Street, 6th Floor, Toronto, ON M5T 3M7, Canada; 9Sechenov First Moscow State Medical University (Sechenov University), Alexander Solzhenitsyn Street 28/1, 109004 Moscow, Russia; 10Institute of Clinical Psychology and Psychotherapy & Center for Clinical Epidemiology and Longitudinal Studies, Technische Universität Dresden, Chemnitzer Str. 46, D-01187 Dresden, Germany

**Keywords:** alcohol, control policies, taxation, availability, marketing ban, Europe, international coordination

## Abstract

Alcohol is a major risk factor for burden of disease. However, there are known effective and cost-effective alcohol control policies that could reduce this burden. Based on reviews, international documents, and contributions to this special issue of International Journal of Environmental Research and Public Health (IJERPH), this article gives an overview of the implementation of such policies in the World Health Organization (WHO) European Region, and of best practices. Overall, there is a great deal of variability in the policies implemented between countries, but two countries, the Russian Federation and Lithuania, have both recently implemented significant increases in alcohol taxation, imposed restrictions on alcohol availability, and imposed bans on the marketing and advertising of alcohol within short time spans. Both countries subsequently saw significant decreases in consumption and all-cause mortality. Adopting the alcohol control policies of these best-practice countries should be considered by other countries. Current challenges for all countries include cross-border shopping, the impact from recent internet-based marketing practices, and international treaties.

## 1. Introduction

Alcohol use is a major contributor to the global burden of disease and injury [1,2,3], and Europe is no exception. On the contrary, in terms of proportional burden of alcohol use (i.e., the proportion of mortality and burden of disease that could be avoided by reducing the use of alcohol), Europe is the region with the highest attributable health burden [3,4], with alcohol-attributable mortality currently at 9.4% (95% confidence interval, CI: 8.6–10.5%) and alcohol-attributable burden of disease at 10.3% (95% CI: 9.9–11.1%; all burden numbers taken from [3] for the European Region of the World Health Organization (WHO)). In other words, more than every tenth year of life lost prematurely in the WHO European Region is due to the use of alcohol. Mortality and morbidity due to the consumption of alcohol stems from four major disease categories: alcohol use disorders, cancer, cardiovascular diseases, and injuries. The high level of alcohol-attributable burden is no surprise, as it is also the region with the highest level of alcohol consumption [5].

As a consequence, one would expect a high level of alcohol control policy activity in this region aimed at reducing consumption levels and attributable burden (for an overview of alcohol control policies in different regions of the world, see [6]). Indeed, the WHO European Region has for many years been at the forefront of policy initiatives to address alcohol consumption, namely the European Charter on Alcohol, adopted by the WHO European Region Member States in 1995 [7], or the European Ministerial Conference on Young People and Alcohol, held in Sweden in 2001 [8], long before the adoption in 2010 of the WHO Global Strategy to Reduce the Harmful Use of Alcohol [9].

However, despite these early initiatives, alcohol control policies in Europe are still diverse, and different combinations of them are applied in each country. The best practices or the most effective policies to reduce alcohol-attributable harm are not necessarily the ones adopted (see below). Consequently, we cannot generalize about alcohol control policies in Europe, or about trends resulting from those policies that would hold true for Europe as a whole. This lack of generalizability extends to any discussion regarding the fact that the WHO European Region is the only region that has already reached the goal of a 10% reduction of harmful use of alcohol, usually defined as adult alcohol per capita consumption [10], as specified in the WHO’s global non-communicable disease action plan [11] (for more information on reaching this goal in the WHO European Region, see [12,13]).

This situation was one of the reasons why a special issue on “Alcohol control policy and health” [14] was initiated by the International Journal of Environmental Research and Public Health (IJERPH). It is the aim of this contribution to: (a) characterize different alcohol control policies in Europe (Point 3.1. below; for background information, see [15,16]), (b) identify best practices (Point 3.2. below; for a general review of effectiveness of alcohol control policies, see [17,18]), and (c) make recommendations for future changes (Point 4.2. below). In doing so, we distinguish three strategies employed by alcohol policies: those directed at the general population, those directed at high-risk drinkers, and background policies that help define the local environment of a particular alcohol policy (see Figure 1).

## 2. Materials and Methods

In order to fulfill the above specified objectives, we conducted a narrative review. As a start, we will define the main terms:**Alcohol control policies** were used as defined by Babor and colleagues [17] as public policies [19] (i.e., authoritative decisions made by governments through laws, rules, and regulations, which pertain to the relation between alcohol, health, and social welfare). The word authoritative indicates that the decisions arose from the legitimate purview of legislators and other public interest group officials, not from private industry or related advocacy groups. It has become customary in the field to distinguish 10 broad categories and areas of policies based on the already mentioned WHO Global Strategy to Reduce the Harmful Use of Alcohol [9]: leadership, awareness, and commitment (usually measured via national alcohol plans); health services response; community action; drinking and driving policies and countermeasures; regulating availability of alcoholic beverages; regulating marketing of alcoholic beverages; pricing policies; reducing the negative consequences of drinking and alcohol intoxication; reducing the public health impact of illicit alcohol and informally produced alcohol; and monitoring and surveillance (for underlying principles in formulating policies, see [20]). Empirically, the so-called “best buys” of the WHO ([21], which comprise increases in price for alcoholic beverages via taxation or other pricing policies, reduction of availability of alcoholic beverages, and bans on advertising and marketing) have been found to be the most effective and cost-effective alcohol policies to reduce health burden [17,22], albeit with different timeframes of impact (taxation and availability tend to include immediate effects on consumption and harm following implementation: [23,24,25]). The three “best buys” have been highlighted in the upper left-hand box of Figure 1.**Europe** was defined here as being equivalent to the WHO European Region, which includes countries that are geographically located in Asia [26]. We chose this definition because alcohol control policies are in part impacted by the WHO regions and, thus, to reach our objectives, it makes sense to use this broader definition.**Best practices** were identified by their proven ability to reduce alcohol-attributable burden of disease and injuries, and reported by their effects on mortality and disability-adjusted life years (see [3]; for underlying concepts and definitions, see [27,28,29]). Obviously, the three “best buys”, if implemented at a sufficiently high level (for instance, taxation increases, which markedly reduce the affordability of alcoholic beverages), would meet the threshold for best practices.

We based our analyses on the contributions [12,20,30,31,32,33,34] to this special issue, [14] on major reviews and intergovernmental publications on alcohol control policies since 2010 (such as [15] or [35]), and on empirical examinations of the impacts of such policies in Europe.

## 3. Results

### 3.1. Alcohol Control Policies in Europe During the Past 10 Years: An Overview

**Taxation and pricing policies:** Overall, of all of the alcohol control policies, the implementation of the “best buys” policies was the weakest, whereas (with the exception of former Soviet Union countries and the Eastern part of the WHO European Region [36]) less effective or limited policies, such as establishing drinking and driving countermeasures or monitoring and surveillance systems, were far more commonly implemented [6,15,37]. Even when the “best buys” policies were implemented, it was not clear whether the policies were implemented as intended in the original definition as a “best buy”, which could reduce alcohol-attributable harm. For example, in the WHO scaling, implementing an excise tax increase already counts as an implementation of a “best buy” [17], even if this increase does not equal the amount lost to inflation.

To elaborate further on this example, if excision or other taxation of alcohol is increased, leading to a price increase, the important economic question is whether such price increases are large enough to cover inflation and salary increases since the previous tax change [23,38], thereby reducing affordability [39]. For instance, a recent report from the WHO also showed that only 16 of 53 (30%) of countries reported that they have alcohol duties (beer, wine, or spirits) linked to inflation, a situation that has likely resulted in increased affordability of alcoholic beverages throughout the region [38].

It has been shown that overall affordability of alcoholic beverages has increased in recent decades; in other words, prices did not keep up with increased income (however, the last overview for the European Union (EU) is dated before 2010: [40]; the only later publication covering most of the countries in the WHO European Region is restricted to beer and shows that, in an overwhelming majority of countries, beer has become more affordable [41]).

The most notable exceptions for a decreased affordability in Europe were Russia and Lithuania, with a combination of increases in taxation and minimum prices for vodka and other alcoholic beverages in Russia [42], and two marked increases in taxation in Lithuania (for an overview of alcohol control measures in Lithuania: [30]; for resulting affordability: [43]), with the first imposed at a time of average salary decreases due to the economic crisis of 2008, leading to lower affordability. The example of Russia included a pricing policy other than taxation: minimum pricing. Even though this policy was originally introduced in Canada [44], it has mostly been applied in the Eastern part of Europe [36], and more recently in Scotland [45]. The rationale for this measure is to establish a floor price for units of pure alcohol in order to prevent access and reduce the affordability of cheap alcohol, and thus decrease the socioeconomic inequalities of alcohol-attributable harms [46,47].

**Changes in availability**: Availability restrictions on alcohol include measures such as systems to regulate production and distribution of alcohol (e.g., by limiting the number of establishments or hours of retail sales) or setting minimum legal drinking ages. The following seemed to be the most commonly used in Europe today: restrictions on opening hours for sales outlets of alcoholic beverages; restrictions on density of alcohol outlets; and restrictions on the people who are allowed to buy or consume alcoholic beverages, or on places where alcohol can be bought or consumed [17]. Overall, in the European Union and the other countries in the Western part of the WHO European Region, availability has been high and has generally remained unchanged over the past 10 years.

Despite the overall picture of high availability, the changes in availability have, in fact, been small. These changes included widening of availability without explicit changes of policy (example: [48]), or local and/or smaller-scale initiatives to restrict availability, such as partial bans on selling alcoholic beverages off-premises two days per week in a Swiss canton [49]. Changes to availability have also been implemented to increase availability in countries where restrictions aimed at protecting young people and children were already in place. For example, Slovenia introduced alcohol availability restrictions in sports facilities, adopting a new act on availability restrictions (ZOPA-A) that restricted sales of alcoholic beverages to those that contained less than 15 volume percent (e.g., beer and wine, no spirits) at sports facilities one hour before an event’s start time and for its duration [50]. Larger-scale changes, such as the liberalization of more restricted availability in the Nordic countries, took place in the first decade of the twenty-first century [51], even though some forms of alcohol monopolies survived.

There were more large-scale reductions in alcohol availability in the Eastern part of the WHO European Region, such as relatively marked decreases in hours of sale or increases in legal purchasing age. To give just one example, in Lithuania, in 2018, off-premise purchases of alcoholic beverages were restricted to between 10 AM and 8 PM from Monday to Saturday, and between 10 AM and 3 PM on Sunday (exceptions were airports, ferries, train stations, and bars; [30,52]). In the same year, the legal age to purchase or consume alcohol was increased from 18 to 20 years old [30,52].

For the whole region, the legislation on availability has been challenged by two events. First, by increases in internet sales (expected to increase steadily as part of a more general trend [53]) and second, by the coronavirus pandemic, which has led to severe restrictions in the on-premise availability of alcohol [54], but also to increases in off-premise availability in many countries that declared alcohol to be an essential good, widened delivery options for restaurants, and loosened restrictions on internet sales [55]. While restrictions on on-premise alcohol sales will very likely be completely removed after the end of the pandemic, arguments are already being made that the increased availability of off-premise alcohol should be made permanent.

**Bans on marketing and advertising**: As for bans on marketing and advertising, no real progress seems to have been made in most countries, especially with respect to the latest forms of social marketing [56,57]. Thus, even in countries that introduced relatively comprehensive bans, such as Lithuania in 2018 [30,52], the social marketing of alcohol does not seem to be controlled in any country. It seems that while alcohol companies, website operators, and the marketing industry proactively adapt to technological changes in digital media, governments lag behind in enacting regulations to address these new challenges [58]. Modern trade agreements also restrict the ability of countries to regulate the marketing activities of the alcohol industry [59].

**Strategies directed at heavy drinkers and people with alcohol use disorders**: Overall, Europe scores high on these strategies [15,16]. Although screening, brief interventions, and some forms of treatment should be more routinely integrated into primary health care [60], and the legal drinking limit for blood alcohol concentration (BAC) could be decreased to at least 0.05 in some countries [15], the European Region seems to be on a good path with respect to these strategies.

**Environmental strategies**: Environmental strategies are popular among governments, as they face no opposition and create the impression that the government cares about alcohol-attributable harm without the potential negative consequences of taxation increases or availability restrictions. In most cases, education is directed at heavy drinkers or against drinking and driving; themes on alcohol as a carcinogen or on the potential consequences of light to moderate drinking are usually avoided, so that in 2020, the majority of the general population still does not associate alcohol consumption with causing cancer [61]. An interesting strategy, especially against unrecorded consumption, is the routine registration of all alcohol (see [33,62] for details on the practice in Russia, which excludes some forms of unrecorded alcohol). Technically, such a registration is feasible and could be easily accomplished.

### 3.2. Identifying Best Practices

We identified best practices based on marked effects on population all-cause mortality and, thus, life expectancy. This can be considered a strict criterion, as most alcohol control measures have tended to be evaluated against more specific criteria, such as traffic deaths for a specific age group or consumption indices (e.g., see the meta-analyses of Wagenaar and colleagues [63,64], or [18]). Therefore, for our definition of best practices, it was necessary to show decreases in all-cause mortality. Based on this criterion, we would like to highlight alcohol control policies in two countries: Russia [65,66] and Lithuania [67]. What can we learn from these best-practice examples?

Alcohol control policies can make a difference even for all-cause mortality in general populations. This is not news for Russia, as alcohol use has already been established as a major determinant for mortality and life expectancy in that country [68,69]. However, the fact that alcohol policies can immediately and permanently improve mortality rates over and above secular trends in high-income Western democracies in the second decade of the twenty-first century has only recently been established.

The best-practice examples seem to be characterized by clusters of integrated policies with emphasis on the “best buys”, as described above. For instance, in 2017/18, in less than one year, Lithuania implemented the following policies: an increase in excise taxation for beer and wine of more than 100%, associated with price increases of 29% and 1%, respectively [43]; an increase in excise taxation of ethyl alcohol of more than 20% (associated with a 14% increase in the price of Lithuanian vodka [43]); an increase in the legal minimum purchasing and drinking age, along with stricter rules of enforcement; and a full ban on TV, radio, and internet advertisements [30,52]. During 2017, all-cause mortality for both sexes and across all ages decreased by 4.8%, over and above secular trends [67]. Similarly, the periods of high density of effective alcohol control policies in short periods of time led to the most marked decreases in all-cause mortality [65,66,70]. Thus, the period of 2010–2013, with measures to increase prices and restrict availability and marketing of alcohol, led to decreases in mortality, over double the number for men (−1.2 per 100,000 per year) as for women (−0.5 per 100,000 per year; [65]). As expected, the changes in alcohol control policy not only led to abrupt changes in trends in all-cause mortality, but also seem to have contributed to longer-term trends.

The impact of alcohol control policies was not limited to all-cause mortality. Other indicators, such as alcohol-attributable traffic crashes or injuries [71] and hospitalizations [65,70], were impacted as well. Consider the following example: between 2004 and 2019, on average, each implemented policy measure in Lithuania permanently reduced the proportion of alcohol-attributable crashes by 0.55%, the proportion of alcohol-attributable injuries by 0.60%, and the proportion of alcohol-attributable deaths by 0.13% [71]. The majority of these measures involved price increases and availability restrictions, and were not primarily intended to impact traffic harms [30,71].

The reductions in the level of alcohol consumption caused by pricing and availability measures, which by definition can only be applied to recorded consumption, were not compensated for by similar increases in unrecorded consumption (for definitions of recorded and unrecorded consumption, see [72,73]; for estimates of recorded and unrecorded consumption in both countries, see [5,67,68]).

## 4. Discussion

### 4.1. Limitations

The following limitations need to be acknowledged. This overview was based on published articles. We therefore may have missed important examples and trends, especially for countries where evidence on policy and its effects are less frequently reported. The reliance on published sources also led to the choice of two countries that were among the most heavily consuming countries a decade ago. However, other countries in the region, such as Kyrgyzstan, used alcohol control policies and markedly reduced their consumption from lower levels, or to keep the consumption level low, such as Turkey ([5], for changes in consumption level; [6] for policies), and of course, alcohol-attributable harm followed suit [3]. We did not report on those countries, as there were fewer publications with detailed evidence available, especially for unrecorded consumption. Turkey appears to have a problem with methanol deaths as part of unrecorded consumption [73,74], independent of the most recent cases in connection to COVID-19 [75]. While this indicates some level of unrecorded consumption, data is sparse as to the magnitude of this problem.

Second, almost all evidence on the effects of alcohol control policies is based on natural experiments and interrupted time-series analyses. Even though such studies provide some control for alternative explanations, we cannot fully rule out the effects of other events occurring at the same time the policy was in effect, impacting alcohol-attributable harm [76]. Third, since all analyses are based on specific historical situations, conclusions may change in light of completely unexpected situations, such as the current coronavirus pandemic [54].

### 4.2. Where Should We Go?

One conclusion is that countries of the region could learn from Lithuania and Russia in terms of alcohol control policies. While such alcohol control policy changes certainly would result in thousands of deaths avoided (for an example in terms of the number of lives saved by increases in taxation, see [77]), a number of things need to be taken into consideration. First, successful alcohol control policies need the support of a large segment of the population (e.g., [32]), and to gain such support, increased alcohol health risk awareness and clear messages about alcohol use and its consequences are necessary, mirroring tobacco awareness-raising campaigns (as part of the Framework Convention [78]). To give but one example, while it has been long known that alcohol use is a major causal factor for cancer [79], leading to 80,000 alcohol-attributable cancer deaths yearly in the European Union alone, as already indicated, this fact is not commonly known to a majority of the general populations of European countries [61]. One way to change this would be via the placement of warning labels on alcoholic beverage containers [80]. While information about the health impact of alcohol use, such as the alcohol-cancer link, seems important to garner support for alcohol policies [81], such information has often been lacking, or the potential beneficial effects of alcohol have been dominant in the public discourse. Empirically, there is also an inverse relationship between drinking in excess of low-risk drinking guidelines and knowledge about the impact of alcohol; people who drink in excess of guidelines, on average, have less knowledge about the detrimental impacts of alcohol use on health [82]. Given these facts, warning labels should be mandatory for alcoholic beverages in European countries, informing consumers about the main health impacts of alcohol consumption. Currently, this is only mandatory in the Eurasian Economic Union [34].

A second point, which is important for inclusion in future alcohol control policies, concerns the establishment of comprehensive approaches. As we have already mentioned, in both best-practice countries (Russia and Lithuania), unrecorded alcohol did not appear to make up for the decreases in recorded consumption as predicted by some stakeholders, such as the alcohol industry. This situation seems to be the result of both countries attempting to address the problem of unrecorded consumption by implementing taxation increases and availability restrictions. For example, Russia did close some of the loopholes for unrecorded consumption by regulating and registering all forms of ethanol, although some untaxed medicinal forms still remain [65,83]. Moreover, Russia has developed what is currently one of the most complete alcohol tracking systems worldwide, the Unified State Automated Information System (EGAIS), to track alcohol production volumes and sales [65,66]. Despite the fact that some level of unrecorded alcohol continues to exist, this form of alcohol did not and does not necessarily have to impede the success of alcohol control policies directed at reducing recorded consumption.

Another aspect of comprehensive alcohol policy concerns the coordination of efforts between neighboring countries. The need for coordination and a shared vision is greater for smaller neighboring countries due to the close proximity of their borders. Lithuania in particular and the Baltic countries in general are good examples. Because of the increases in price in Lithuania, and the resulting increase in the price differential with Latvia, cross-border shopping increased, similar to an increase in cross-border shopping between Latvia and Estonia [84]. Instead of harmonizing taxation policies and thus prices between close neighbors in a concerted action, countries seemingly try to solve the problem with unilateral actions, such as taxation decreases, followed by similar decreases in other countries. Clearly, good alcohol control policies will need to take into consideration the economic environment, and the “invisible hand of the market” in such situations will not lead to the best solutions for the common good [85].

More international integration and collaboration is also necessary for the next round of challenges for alcohol control policies: the increase of availability through online sales (see above) and the globalization of marketing on social media for alcoholic beverages, which is expected to increase rapidly [56,57]. Not only will national governments, especially governments of smaller countries, likely be overwhelmed by regulating global markets, but, if current trends in globalization continue, national regulation may no longer be possible for at least some aspects of it [59]. In this situation, the possibility of adopting strong legal instruments could be considered on the international level. Different kinds of such instruments have been suggested, including an international convention, similar to the Framework Convention on Tobacco [78]. However, these legal instruments do not seem to be in sight at the global level, as the proposal from low- and middle-income countries to consider a working group “to review and propose the feasibility of developing an international instrument for alcohol control” was turned down at the WHO Executive Board earlier this year ([86]; see also [87]). At the same time, while such legal instruments could be important tools for international coordination and cooperation, domestic legislation is required to implement any international obligations, requiring political buy-in from Member States, regardless of the legal status of an international instrument [88]. At this point, while we see some buy-in from low- and middle-income countries for international treaties, high-income countries, and in particular the WHO European Region Members, seem to be more open to strengthening national policy implementation under a new Framework for Action to strengthen the implementation of the WHO European Action Plan to Reduce Harmful Use of Alcohol.

Another key player in alcohol control for parts of Europe is the EU. The 2015 Council Conclusions on “an EU strategy on the reduction of alcohol-related harm”, or the 2017 Opinion of the European Committee of the Regions on “the need for and way towards an EU strategy on alcohol-related issues”, set clear evidence-based directions for the way forward in the development and implementation of a sound alcohol policy across the European Region [89,90]. Several Member States, the European Parliament, and stakeholders in public health have been encouraging bolder steps towards alcohol control over the EU [91], supported by the conclusions of various evaluations, such as the assessment of the added value of EU strategy to support Member States in reducing alcohol-related harm [92] or the progress evaluation report on the Action Plan on Youth Drinking and on Heavy Episodic Drinking (Binge Drinking) 2014–2016 [93]. However, the development of a new comprehensive alcohol strategy for the EU is still pending. Some factors are hindering progress, such as efforts by the alcohol industry and of some Member States to actively lobby against a unified alcohol policy, which could be detrimental to their economic activities [94]. One consequence is that there are no EU regulations regarding either the placement of warning labels or the text printed on them on alcoholic beverages, which contrasts with the regulations established for any ordinary non-alcoholic commodity on the EU market today, such as those which require the labeling of milk cartons [95]. Such regulations would be an easy win, and have already been adopted within the Eurasian Economic Union [36,80]. The major challenge for the EU today, and a major hope in terms of reduction of alcohol-related harm across Europe, is to respond to the recent changes in alcohol marketing practices and to help Member State governments coordinate policy responses, thereby helping to establish a truly comprehensive European alcohol control policy [91].

## 5. Conclusions

Best practices in alcohol control policy in some European countries have been shown to save lives and increase the life expectancy of national populations. Such practices need to be considered across Europe. However, formulating and implementing alcohol control policies is a challenging task, one which requires securing the support of the general population and answering in a proactive, innovative, and coordinated manner to key challenges, such as those resulting from the new forms of alcohol delivery and digital marketing, which have resulted from recent rapid technological change.

## Figures and Tables

**Figure 1 ijerph-17-08162-f001:**
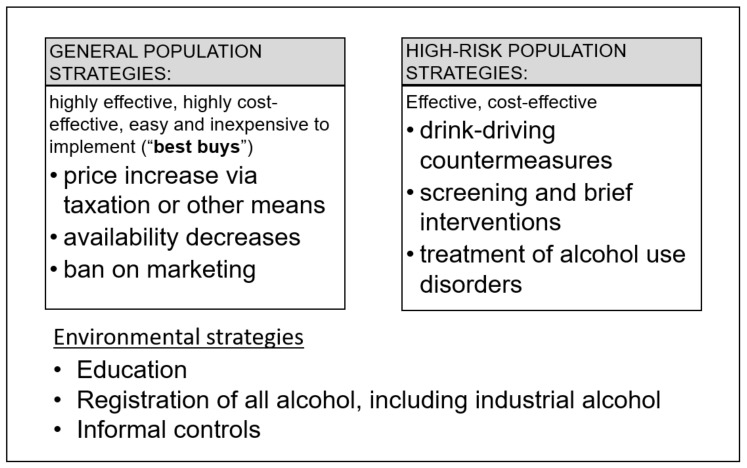
Overview of major strategies employed in alcohol control policies.

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
