# Peer review of "Alcohol Control Policy in Europe: Overview and Exemplary Countries"

_ijerph, 2020, doi:10.3390/ijerph17218162_

Round 1

Reviewer 1 Report

Alcoholism is one of the most common problems in the modern world. As defined by the World Health Organization: Alcoholism is any form of drinking that exceeds traditional and customary consumption, or community-wide, social drinking, regardless of what may lead to it. It is a disease that develops slowly and very insidiously. It is commonly said that alcoholism is the loss of control over the consumption of alcoholic beverages, which can lead to death. Alcoholism is also a primary disease, which means that it underlies many other somatic and mental illnesses. The topic presented in the manuscript is very important and is of particular importance in the light of alcoholism prevention.

The manuscript is written in accordance with the journal's requirements, in a grammatically correct and logical manner. It is a good source of information and points out any factors that need to be changed in alcohol policy. Therefore, it should be published only after the correction of key words. The manuscript contains too many keywords. this should be corrected, leaving only those that best reflect the subject of the manuscript.

Author Response

Alcoholism is one of the most common problems in the modern world. As defined by the World Health Organization: Alcoholism is any form of drinking that exceeds traditional and customary consumption, or community-wide, social drinking, regardless of what may lead to it. It is a disease that develops slowly and very insidiously. It is commonly said that alcoholism is the loss of control over the consumption of alcoholic beverages, which can lead to death. Alcoholism is also a primary disease, which means that it underlies many other somatic and mental illnesses. The topic presented in the manuscript is very important and is of particular importance in the light of alcoholism prevention.

Thank you.  We have added some background on alcohol dependence and alcohol use disorders to the manuscript.

The manuscript is written in accordance with the journal's requirements, in a grammatically correct and logical manner. It is a good source of information and points out any factors that need to be changed in alcohol policy. Therefore, it should be published only after the correction of key words. The manuscript contains too many keywords. this should be corrected, leaving only those that best reflect the subject of the manuscript.

We reduced the number of keywords.

Regarding language, the manuscript was edited by a professional copy editor.

Reviewer 2 Report

The Author present a study on the European policy regarding the alcohol control.

The subject deserves interest and attention, nevertheless, the manuscript has some serious flaws.

First of all, the manuscript should be considered as minireview and not as research article. Basically, it consists in a report about the current policy in Russia and some other east Europe Countries, which a marginal paragraph, in the last part of the manuscript, with some commentaries on the European Union policy.

This last observation open to another question: the manuscript is not referred to the European policy. It should be clearly indicated that this overview is referred to Russia and Lithuania.

Also, no figure or scheme has been reported. The reading without any graphic element results difficult.

Author Response

The Author present a study on the European policy regarding the alcohol control.

The subject deserves interest and attention, nevertheless, the manuscript has some serious flaws.

First of all, the manuscript should be considered as minireview and not as research article. Basically, it consists in a report about the current policy in Russia and some other east Europe Countries, which a marginal paragraph, in the last part of the manuscript, with some commentaries on the European Union policy.

We have changed the article type and resubmitted the manuscript as a review.  Policies throughout Europe are included, and the scope is better explained now.  It just happens that most exemplary alcohol control policies, as defined by the WHO’s best buys, have been implemented in the East.  However, we also report on Scotland and other countries with exemplary alcohol policies.

This last observation open to another question: the manuscript is not referred to the European policy. It should be clearly indicated that this overview is referred to Russia and Lithuania.

As indicated above, Russia and Lithuania are just highlighted as countries which implemented all of the best buys in recent years.  This is now clarified in the revised article.  The new title, based on the changes described above, is:

Alcohol control policy in Europe: overview and exemplary countries

Also, no figure or scheme has been reported. The reading without any graphic element results difficult.

We have introduced a Text Figure and more subheadings to guide the reader through the material.

Regarding English, the manuscript has been edited by a professional editor.

Reviewer 3 Report

This was an interesting overview of recent changes in European alcohol policy. This paper described various alcohol policies, which policies were considered ‘best buys’, metrics that would be impacted by the policies, and the overall direction of expected/observed change in those metrics.

Overall, this was a well written paper which delivered important public health action items in the “Where Should we Go?” section. My two main comments for improving this manuscript are that:

  1. The structure of the paper should be more clearly outlined in the last paragraph of the introduction.

I think it’d be clearer for the reader if the explicit points made in the limitations section that there was no specific primary data presented, and the review was narrative rather than systematic, were placed earlier in the manuscript.

  1. This paper served the function of an overview well – it was like an index on how various studies ‘slot’ into the literature. However, I would like to see a few more specific examples of the scale of the effects described as “abrupt changes” and “marked decreases”.

For example, on P5. L 199 the authors mention that the primary measure they were considering was all-cause mortality, and that there were “marked decreases in all-cause mortality [61, 62, 66]”. It would be useful for the reader to have a summary of what these three referenced studies found in terms of specific/average number of years decreased. If one of the premises of the paper is to showcase countries with best practice, then I believe having some estimated effects would be more compelling for the reader.

Similarly, there is insufficient detail in the sentence on P5 L204. “alcohol-attributable traffic crashes or injury [67] or hospitalizations [61, 205 66], were impacted as well”, to confirm event that this impact was an improvement/the effects occurred in the expected direction.

Author Response

This was an interesting overview of recent changes in European alcohol policy. This paper described various alcohol policies, which policies were considered ‘best buys’, metrics that would be impacted by the policies, and the overall direction of expected/observed change in those metrics.

Overall, this was a well written paper which delivered important public health action items in the “Where Should we Go?” section. My two main comments for improving this manuscript are that:

1. The structure of the paper should be more clearly outlined in the last paragraph of the introduction.

I think it’d be clearer for the reader if the explicit points made in the limitations section that there was no specific primary data presented, and the review was narrative rather than systematic, were placed earlier in the manuscript.

We have provided an overview of the structure, and further guided the reader with an overview Figure and more subheadings.

2. This paper served the function of an overview well – it was like an index on how various studies ‘slot’ into the literature. However, I would like to see a few more specific examples of the scale of the effects described as “abrupt changes” and “marked decreases”.

We have provided a number of more specific examples throughout the text.

For example, on P5. L 199 the authors mention that the primary measure they were considering was all-cause mortality, and that there were “marked decreases in all-cause mortality [61, 62, 66]”. It would be useful for the reader to have a summary of what these three referenced studies found in terms of specific/average number of years decreased. If one of the premises of the paper is to showcase countries with best practice, then I believe having some estimated effects would be more compelling for the reader.

We have given specific quantifications for the decreases in all cause-mortality in the revised version.

Round 2

Reviewer 2 Report

All the issues have been addressed.